# Comparison of Train of Four Measurements with Kinemyography NMT DATEX and Accelerography TOFscan

**DOI:** 10.3390/medsci9020021

**Published:** 2021-03-29

**Authors:** Cyrus Motamed, Migena Demiri, Nora Colegrave

**Affiliations:** Institut Gustave Roussy Rue Camille Desmoulins, CEDEX, 94080 Villejuif, France; migena.demiri@gustaveroussy.fr (M.D.); nora.colegrave@gustaveroussy.fr (N.C.)

**Keywords:** TOFscan, kinemyography, neuromuscular monitoring, neuromuscular blockade

## Abstract

Introduction: This study was designed to compare the Datex neuromuscular transmission (NMT) kinemyography (NMTK) device with the TOFscan (TS) accelerometer during the onset and recovery of neuromuscular blockade. Patients and methods: This prospective study included adult patients who were scheduled to undergo elective surgery with general anesthesia and orotracheal intubation. The TS accelerometer was randomly placed at the adductor pollicis on one hand, and the NMTK was placed on the opposite arm. Anesthesia was initiated with remifentanil target-controlled infusion (TCI) and 2.0–3.0 mg/kg of propofol. Thereafter, 0.5 mg/kg of atracurium or 0.6 mg/kg of rocuronium was injected. If needed, additional neuromuscular blocking agents were administered to facilitate surgery. First, we recorded the train of four (TOF) response at the onset of neuromuscular blockade to reach a TOF count of 0. Second, we recorded the TOF response at the recovery of neuromuscular blockade to obtain a T4/T1 90% by both TS and NMTK. Results: There were 32 patients, aged 38–83 years, with the American Society of Anesthesiologists (ASA) Physical Status Classification I–III included and analyzed. Surgery was abdominal, gynecologic, or head and neck. The Bland and Altman analysis for obtaining zero responses during the onset showed a bias (mean) of 2.7 s (delay) of TS in comparison to NMTK, with an upper/lower limit of agreement of [104; −109 s] and a bias of 36 s of TS in comparison to NMTK, with an upper/lower limit of agreement of [−21.8, −23.1 min] during recovery (T4/T1 > 90%). Conclusions: Under the conditions of the present study, the two devices are not interchangeable. Clinical decisions for deep neuromuscular blockade should be made cautiously, as both devices appear less accurate with significant variability.

## 1. Introduction

Neuromuscular monitoring is mandatory before recovery to prevent residual paralysis and its complications. In addition, monitoring helps to avoid inadequate dosing because of inter-individual and intra-operative variability [1] and other critical complications [2,3].

Several monitoring sites, such as the adductor pollicis muscle, orbicularis occuli, supercilii [4], or posterior tibial nerve (plantar flexor muscle) [5,6,7], are available; however, the adductor pollicis is the main site in which objective quantitative monitoring can be performed [1]. This latter site may not be available for quantitative measurement with positioning during surgery or other procedures that require general anesthesia and muscle relaxation, such as procedures for orthopedic, cervico facial surgery, or interventional radiology, in which both arms may not be available to position sensors to detect muscle relaxation.

We previously compared the TOFscan (TS) accelerometer to the TOF Watch accelerometer [8]. The TS seemed to have some advantages, including the absence of the need for calibration without affecting the recovery results. However, the TS accelerometer has not been compared to another method of monitoring or a device that has a similar external shape, such as the neuromuscular transmission kinemyography (NMTK) device, for which we also previously reported reasonable accuracy in clinical practice [9,10].

Both of these devices are easier to install in comparison to other devices that need the hand to be immobilized to avoid the movement of the thumb altered by other fingers.

This study was designed to assess the interchangeability of these two devices (TS and NMTK). Both devices have the same external sensor shape, which can be employed in other positions than the classic extended arm, and placed at the wrist to stimulate the peripheral nerve of the adductor pollicis muscle during the onset and recovery of neuromuscular blockade (NMB).

## 2. Patients and Methods

This prospective and observational study was approved by our hospital’s ethical committee (avis n° 2013 A00967) and institutional review board in June 2013. Informed consent was obtained from all patients.

This study comprised the second phase of the initial study that compared the TS accelerometer to the TOF Watch accelerometer, and was conducted from January to October 2018 [8].

We included adults who underwent general anesthesia with orotracheal intubation facilitated by neuromuscular blocking drug administration, and who had both of their forearms free to permit the monitoring of neuromuscular blockade via the adductor pollicis muscle.

Patients who were younger than 18, pregnant, breastfeeding, had an allergy to neuromuscular blocking agents, anticipated to have difficult intubation and programmed to be intubated without muscle relaxant (according to the anesthetist in charge), and patients whose arms could not both be used for neuromuscular monitoring were excluded from the study.

The patient was placed in a supine position before the insertion of an intravenous line and the administration of an electrocardiogram (EKG), pulse oximetry, and noninvasive blood pressure. The forelimbs were extended passively and the antebrachium was supported along its length in a horizontal position. Devices were installed on each hand on a random side and anesthesia was initiated with remifentanil target-controlled infusion (TCI), followed by 2.0–3.0 mg/kg of propofol.

After loss of consciousness, both devices were simultaneously turned on. While the NMTK had an autocalibration procedure, the TS started monitoring without calibration. A repetitive train of four (TOF) stimulation with an intensity of 50 mA every 12–15 s was therefore initiated.

Thereafter, 0.5 mg/kg of atracurium or 0.6 mg/kg of rocuronium was injected until a 0 response was obtained at the TOF before tracheal intubation, which was performed arbitrarily when we obtained no further response from the NMTK, as it was considered our first method of monitoring. During surgery, anesthesia was maintained with an inhalational agent, and remifentanil was administered with a target-controlled infusion mode (Base Primea, Fresenius^®^, Fresnes, France). Ventilation was controlled, and end-tidal carbon dioxide tension (ETCO2) was maintained between 32 mmHg and 36 mmHg. Repeated boluses or intravenous infusion of neuromuscular blockade agent were administered to obtain TOF responses between 0 and 2, if necessary for surgery.

At the end of surgery, if needed, reversal of neuromuscular paralysis was performed with neostigmine in combination with atropine or sugammadex, based on the results of the NMTK T4/T1 (TOF ratio) > 90%, and extubation was performed accordingly in conjunction with other clinical parameters, such as awakening.

The following data were recorded: -For the onset of neuromuscular blockade: times, in seconds, to obtain T4/T1 < 70%, 50%, 10%, and 1%, a TOF count < 4, and a TOF count of 0.-For the recovery of blockade: times, in minutes, to obtain 1, 2, 3, and 4 responses at the TOF, and then to obtain T4/T1 > 10%, 25%, 40%, 70%, 75%, and 90%.

The sample size was chosen in accordance with our previous investigation concerning TS [8] and other similar studies [9,11,12,13,14,15]. We already compared the TS and the TOF Watch accelerometer [8]; in this study, 32 patients were included and a mean bias of 26 s was observed between the two types of accelerometers. We thought by maintaining a similar sample size, this difference would be greater since, in the current study, we had two different methods of monitoring.

Bland–Altman analysis was performed for the onset of the 0 response and a TOF ratio recovery to 90%. The NMTK, which was the oldest method, was considered the first method, and the TS (the newer method) was considered the second method.

Once the scatter plot, bias, and limits of agreement were displayed, the software also gave additional information, including the distribution of data by performing the Wilk–Shapiro distribution test, with a *p*-value of less than 0.05 rejecting the normality. In addition, a regression line with a 95% confidence interval and its equation were calculated and incorporated into the plot.

We hypothesized that a bias of more than 10% of the difference and or/limits of agreement of higher than 20% of the bias would rule out interchangeability in this context, which requires a swift clinical decision.

The mean or median time differences between the devices were also compared with the Wilcoxon rank sum test or Student’s *t*-test (paired samples), depending on the distribution of data verified visually by a histogram, and followed by the Wilk–Shapiro test. The differences were considered statistically significant when the *p*-value was less than 0.05, with a power of 80%. Medcalc V15.8 Ostend, Belgium was employed for the calculations.

The results are expressed as the mean ± standard deviation (SD), if normally distributed, or as the median (25–75) percentiles otherwise.

## 3. Results

There were 32 patients enrolled in the study, of which 2 patients were excluded from the data analysis because of insufficient data and a calibration error in the NMTK. Thus, 30 (American Society of Anesthesiologists (ASA) I–II, 5–25) were included in the data analysis.

The patients were aged 56 ± 12 years; their body mass index range was 24.9 ± 5 kg·m^−2^. All patients had general surgery with both forearms free.

Atracurium was used in 27 patients, and rocuronium was administered in 3 patients. A total of 17 patients received a single injection of neuromuscular blockers for intubation, and 13 patients received additional doses. No infusion was employed. In 13 patients, a pharmacological reversal was administered (neostigmine and atropine for 11 patients, and sugammadex for 2 patients) before emergence from anesthesia.

The mean supramaximal stimulation current for the NMTK was 38.1 ± 12 mA. The TS accelerometer does not perform calibration. A total of 455 paired responses were recorded.

The comparison for the onset of neuromuscular blockade is displayed in Table 1, Figure 1 displays the Bland and Altman representation (difference plot), revealing a bias of 2.7 s (earlier onset for NMTK) and an upper/lower limit of agreement of +109.7/−104.3 s. The regression line for the onset of blockade suggests a proportional bias increasing with time to reach full neuromuscular blockade (Figure 1). Table 2 displays statistical information of the graph.

For recovery, the mean first twitch response for the NMTK appeared 10 min before TS (*p* = 0.0001), and the second twitch for the NMTK appeared 7 min before TS (*p* < 0.0005). The recovery from deep neuromuscular blockade (TOF < 2) was detected more quickly by NMTK than TS (*p* < 0.001) (Table 3).

However, the overall difference was not significant for moderate or total neuromuscular blockade recovery (T1/T4 > 90%), with a bias of 0.6 min (36 s) and an upper/lower agreement of 21.8/−23.1 min, which rules out interchangeability. The data and the Bland and Altman ratio plot are presented in Figure 2. Table 4 displays statistical information of the graph.

Table 1 and Table 3 display the values at each time point for the onset and recovery of neuromuscular blockade.

## 4. Discussion

This study showed no statistically significant differences between the two devices during the onset and late recovery of neuromuscular blockade. However, wide limits of agreements during both onset times and late recovery rule out interchangeability between the devices.

Quantitative monitoring of neuromuscular blockade is now fully recommended in every clinical situation, including during surgery, to avoid over or under dosage [1].

Appropriate monitoring is best achieved when monitoring begins during the onset and continues until the end, with initial proper calibration when necessary for the device [16].

Our study was conducted in accordance with the study of Ezer et al. [17]. In their study, kinemyography and accelerography were considered interchangeable. Our study was different since we assessed induction and recovery, as well as the use of the TS accelerometer, which has approximately the same physical shape as a kinemyography sensor, although calibration is not necessary, which might partly explain the difference between the two studies.

A clinical implication from the present study for the onset of neuromuscular blockade is that waiting until a 0 response after TOF stimulation is probably too long, especially when a TOFscan is used, which confirms old investigations that laryngeal muscles are paralyzed earlier than adductor pollicis [18].

This study also implies caution is necessary during recovery, before swift decisions being made with deep neuromuscular blockade (less than four responses), as the variability of the results increases with both devices.

Objective monitoring is only possible via the adductor pollicis because of the lack of commercial monitors for other sites, such as the corrugator supercilii or orbicularis oculi [4]. The values described in these studies were only included for research since a new device had not been elaborated and commercialized.

However, for quantitative adductor pollicis monitoring, several techniques, including accelerography, kinemyography, and recently, compressomygraphy and electromyography, are commercially available.

We believe the classic extended arm in the supine position to monitor neuromuscular blockade is no longer required or possible for a high percentage of surgical or interventional or diagnostic procedures. Therefore, monitoring these positions could be problematic since most monitors need forearm fixation to permit free movement of the thumb.

In routine clinical practice, we suggest that the TS accelerometer might be more appropriate in a nonconventional operation room for procedures such as interventional radiology gastrointestinal endoscopy, and brachytherapy or in non-supine positions (orthopedic surgery) where forearms are not extended. For these specific position requirements, the initial calibration might be compromised after changing the position. Therefore, values might not be adequately reliable mainly during the procedure, and most importantly, for recovery. We previously demonstrated that the absence of calibration will not affect the final results of the TS accelerometer in other non-classic operating room situations. Additionally, the TOFscan uses a three-dimensional accelerometer, which probably helps in this context [8].

There are a few shortcomings of this study. First, with a sample size of only 32 patients, we cannot rule out the lack of power in our results. In the present study, the differences between the devices were clinically and statistically important at a few time points and would have resulted in different clinical decisions, but because of a lack of power for the overall study, we cannot generalize these results. Moreover, as an observational study in current clinical care, we did not change our practice. In addition, the duration of anesthesia, non-depolarizing neuromuscular blocking agent (NMBA) delivery schemes, and temperature control were not standardized. Finally, there was no randomization concerning the dominant arm and the side of each device. This issue has been previously reported [13,14] with no differences between arms, using either accelerography or mechanomyography. Finally, kinemyography is not considered a gold standard monitoring technique [9].

## 5. Conclusions

Both kinemyography and accelerography have similar time profiles for the onset and 90% recovery of neuromuscular blockade; the first and second TOF response appears sooner for the NMTK. Both methods are suitable for positions other than the supine position, with the practical advantage belonging to the TS accelerometer, which does not need a mandatory initial calibration procedure. However, because of the wide limits of agreements, the TS accelerometer and NMTK cannot be used interchangeably.

## Figures and Tables

**Figure 1 medsci-09-00021-f001:**
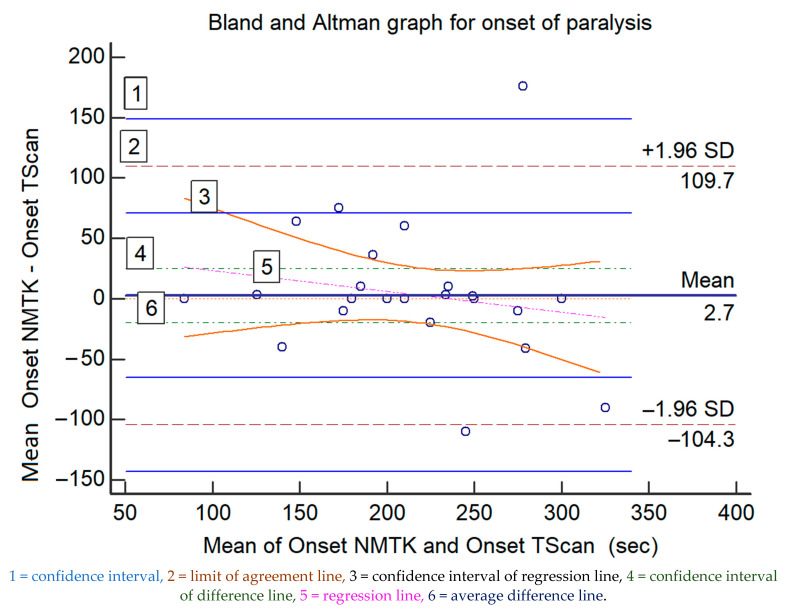
The Bland and Altman representations of the paired measured data for the onset of muscle relaxation (in seconds) of both devices; the values of the *x*-axis represent the mean values of the devices and those of the *y*-axis represent the mean differences between each device.

**Figure 2 medsci-09-00021-f002:**
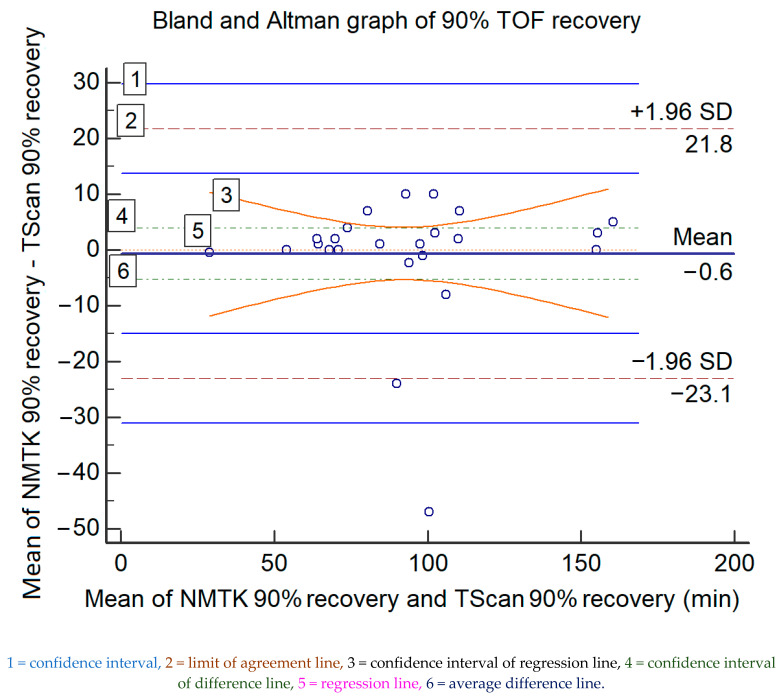
The Bland and Altman representations of the paired measured data for 90% TOF recovery (in minutes) of both devices; the values of the *x*-axis represent the mean values of the devices and those of the *y*-axis represent the mean differences between each device.

**Table 1 medsci-09-00021-t001:** Onset of neuromuscular blockade (s).

	TS	NMTK	*p*-Value
Time to T4/T1 < 70% (s)	149 (110–180)	131 (95–158)	0.6
Time to T4/T1 < 50% (s)	160 (110–195)	159 (125–185)	0.22
Time to T4/T1 < 10% (s)	188 ± 62	169 ± 48	0.2
Time to 1 ≤ TOF count < 4 (s)	190 ± 63	193 ± 56	0.8
Time to TOF count = 0 (s)	216 ± 61	217 ± 69	0.9

Data are presented as mean (s) ± SD or median (s) (25–75). Percentiles are as appropriate with regard to distribution.

**Table 2 medsci-09-00021-t002:** Statistical information of the onset graph.

Arithmetic mean	2.7600
95% CI	−27.8361 to 14.3161
*p* (H_0_: Mean = 0)	0.5143 Wilk–Shapiro test for normality passed
Standard deviation	51.0590
Lower limit	−104.3357
95% Cl	−143.3457 to −70.3257
Upper limit	109.7
95% Cl	56.8057 to 129.8257
Regression Equation	y = −19.3570 + 0.05958x
95% CI	−106.2534 to 67.5393

**Table 3 medsci-09-00021-t003:** Train of four (TOF) recovery from 1 response to 90% (min).

	TS	NMTK	Significance
Time to first response TOF	51 (28–160)	41 (18–130)	0.0001
Time to second response TOF	56 (36–169)	49 (19–160)	0.0004
Time to third response TOF	68 ± 24	55 ± 21	0.003
Time to fourth response TOF	74 ± 34	70 ± 34	0.6
Time to 10% TOF ratio	78 ± 37	77 ± 36	0.9
Time to 25% TOF ratio	81 ± 35	95 ± 86	0.4
Time to 50% TOF ratio	99 ± 86	98 ± 82	1
Time to 70% TOF ratio	107 ± 88	107 ± 66	1
Time to 75% TOF ratio	108 ± 81	109 ± 88	0.9
Time to 90% TOF ratio	116 ± 85	112 ± 83	0.8

Data are presented as mean (min) ± SD or median (min) (25–75). Percentiles are as appropriate with regard to distribution.

**Table 4 medsci-09-00021-t004:** Statistical information of the 90% recovery graph.

Arithmetic mean	−0.6
95% CI	−4.3022 to 3.9936
*p* (H_0_: Mean = 0)	0.9398 Wilk–Shapiro test for normality (passed)
Standard deviation	109.047
Lower limit	−232.275
95% CI	−28.6992 to −14.3558
Upper limit	21.8189
95% CI	14.0472 to 28.3906
Regression Equation	y = 0.296 + (−0.004945)x
95% CI	−13.8276 to 14.4197

## Data Availability

Data available on demand.

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
