# Peer review of "Comparison of Train of Four Measurements with Kinemyography NMT DATEX and Accelerography TOFscan"

_medsci, 2021, doi:10.3390/medsci9020021_

Round 1

Reviewer 1 Report

Thank you for giving me the opportunity to review the manuscript "Comparison of Train of four measurements with kinemyography NMT DATEX and accelerography TOF scan". I read this article with great interest. In the study, the researchers compared two monitors of neuromuscular blockade (the patient is his own control). The authors concluded that the two monitors are interchangeable, except for the first phases of the recovery.

The study is easy to understand, includes methodological issues that could make it exposed to criticism, and lacks originality.

Here are my comments :

  • ABSTRACT

“ Second, we recorded the TOF response at onset of …”,  We should replace “onset” with “recovery”

  • INTRODUCTION

I suggest deleting the first sentence.

“The practice of anesthesiology has expanded to locations other than the traditional surgical operating room (such as interventional radiology), which sometimes necessitates positions different than the traditional straight supine position with arms extended at 90 degrees.”, I suggest deleting this sentence as well, which seems repetitive (see above).

  • PATIENTS AND METHODS

Please justify excluding patients who may be difficult to ventilate or intubate (NMBs facilitate the ventilation in most cases and also the intubation). It would possibly be more appropriate to state that patients induced and intubated without NMB (according to the anesthetist in charge) were excluded.

"The patient was placed in a supine position for surgery ..", the patient is placed in the supine position before the installation of the intravenous line and the monitoring usually, which does not seem to be the case in the text.

The reversal of NMBs is not standardized, it can be performed by neostigmine or Suggamadex. The latter acts much faster. It would have been better to use a single drug. Also, several patients did not receive a reverse. I think the analysis of the recovery includes several elements of heterogeneity.

“The sample size was chosen in accordance with our previous investigation concerning TS (8). Sample size was chosen in accordance with our previous investigation ”. This sentence is repeated.

Even if the sample size is based on previous studies, please explain it. Is it based on a particular calculation? Or did you choose the same number and in this case please justify your choice (Especially since the sample seems limited).

Hope this will help.

Author Response

Reviewer 1

Thank you for giving me the opportunity to review the manuscript "Comparison of Train of four measurements with kinemyography NMT DATEX and accelerography TOF scan". I read this article with great interest. In the study, the researchers compared two monitors of neuromuscular blockade (the patient is his own control)rst phases of the recovery.

The study is easy to understand, includes methodological issues that could make it exposed to criticism, and lacks originality.

Here are my comments :

  • ABSTRACT

“ Second, we recorded the TOF response at onset of …”,  We should replace “onset” with “recovery”

We apologize for this error 

Second, we recorded the TOF response at recovery of neuromuscular blockade to obtain T4/T1 > 10% to 90% by both TS and NMTK

  • INTRODUCTION

I suggest deleting the first sentence.

The sentence has been deleted

“The practice of anesthesiology has expanded to locations other than the traditional surgical operating room (such as interventional radiology), which sometimes necessitates positions different than the traditional straight supine position with arms extended at 90 degrees.”, I suggest deleting this sentence as well, which seems repetitive (see above).

This sentence has been deleted as well

  • PATIENTS AND METHODS

Please justify excluding patients who may be difficult to ventilate or intubate (NMBs facilitate the ventilation in most cases and also the intubation). It would possibly be more appropriate to state that patients induced and intubated without NMB (according to the anesthetist in charge) were excluded.

We changed the sentence as requested :

Patients who were younger than 18, pregnant, with an allergy to neuromuscular blocking agents, patients induced and intubated without NMB (according to the anesthetist in charge) and patients in which  both arms could not be used for neuromuscular monitoring were excluded from the study.

"The patient was placed in a supine position for surgery ..", the patient is placed in the supine position before the installation of the intravenous line and the monitoring usually, which does not seem to be the  case

We corrected that sentence as well. 

The patient was placed in a supine position before insertion of an intravenous line and installation electrocardiogram (EKG), pulse oximetry, and noninvasive blood pressure case in the text.

The reversal of NMBs is not standardized, it can be performed by neostigmine or Suggamadex. The latter acts much faster. It would have been better to use a single drug. Also, several patients did not receive a reverse. I think the analysis of the recovery includes several elements of heterogeneity.

We agree that the possibility of three types of recovery is heterogeneous in term of recovery however this study was designed to compare two devices and the heterogeneity is the same for both devices, in and there was only 3 patients who had rocuronium.

Moreover, as an observational study, in current clinical care we did not change our practice as a consequence there is heterogeneity of practice possibly affecting recovery, however it affects both devices

“The sample size was chosen in accordance with our previous investigation concerning TS (8). Sample size was chosen in accordance with our previous investigation ”. This sentence is repeated.

We removed the repetition 

Even if the sample size is based on previous studies, please explain it. Is it based on a particular calculation? Or did you choose the same number and in this case please justify your choice (Especially since the sample seems limited).

Thank you for your suggestion we added a special paragraph for the sample size 

The sample size was chosen in accordance with our previous investigation concerning TS (8).  In this study thrity two patients were included and a mean bias of 26 sec was observed between two types of accelerometers  for tof recovery , we thought by  maintaining a similar sample size, this difference would be greater since in the current study we had two different methods of monitoring . 

It should also be noticed we found several other studies using similar limited sample size as ours cited below 

  1.     Chau I, Horn K, Dullenkopf A. Neuromuscular monitoring during modified rapid sequence induction: A comparison of TOF-Cuff(R) and TOF-Scan(R). Australas Emerg Care. 2020;23(4):217-20.
  2.     Claudius C, Skovgaard LT, Viby-Mogensen J. Arm-to-arm variation when evaluating neuromuscular block: an analysis of the precision and the bias and agreement between arms when using mechanomyography or acceleromyography. Br J Anaesth. 2010;105(3):310-7.
  3.     Colegrave N, Billard V, Motamed C, Bourgain JL. Comparison of the TOF-Scan acceleromyograph to TOF-Watch SX: Influence of calibration. Anaesth Crit Care Pain Med. 2016;35(3):223-7.
  4.     Demiri M, Colegrave N, Motamed C, Billard V. Comparison of the four-train measurement with a new TOF Cuff(R) device versus TOF Watch(R) accelerometer. Anaesth Crit Care Pain Med. 2020;39(6):891-2.
  5.     Dullenkopf A, Horn K, Steurer MP, Hess F, Welter J. Placement of TOF-Cuff(R) on the lower leg for neuromuscular and blood pressure monitoring during anesthetic induction for shoulder surgeries. J Anesth. 2020;34(1):79-85.
  6.     Motamed C, Kirov K, Combes X, Duvaldestin P. Comparison between the Datex-Ohmeda M-NMT module and a force-displacement transducer for monitoring neuromuscular blockade. Eur J Anaesthesiol. 2003;20(6):467-9.
  7.     Sfeir Machado E, Keli-Barcelos G, Dupuis-Lozeron E, Tramer MR, Czarnetzki C. Assessment of spontaneous neuromuscular recovery: A comparison of the TOF-Cuff((R)) with the TOF Watch SX((R)). Acta Anaesthesiol Scand. 2020;64(2):173-9.

Reviewer 2 Report

Dear authors
I read with interest in your paper, and I appreciate the topics and study design. But I noted some mistakes in statistical analysis.
First, you decided to use a statistics test without providing data about normal distribution for continuous variables. I suggest to perform the Shapiro-Wilk test to find if your data were or not normally distributed, and as a consequence reported as mean and standard deviation or median and interquartile range.
Second, for the correct application of B&A analysis, I suggest reading the following articles: https://pubmed.ncbi.nlm.nih.gov/26110027/ and 10.1016/j.tjem.2018.09.001.
As a consequence, for the lack of statistical analysis, the present article requires major revision.

Author Response

Dear authors

I read with interest in your paper, and I appreciate the topics and study design. But I noted some mistakes in statistical analysis.

First, you decided to use a statistics test without providing data about normal distribution for continuous variables. I suggest to perform the Shapiro-Wilk test to find if your data were or not normally distributed, and as a consequence reported as mean and standard deviation or median and interquartile range.

Thank you for your suggestion 

We performed visual histogram analysis followed by  shapiro wilk for normality distribution test  for all data when comparing time point differences The datAare now represented accordingly.

 In the case of Bland and Altman presentation , we hypothesized an initial estimation of be an  acceptable limit of agreements  in the context of medical device comparison ( the Bias of less than 10% and or  no limit of agreement trespassing 20%) would be appropriate. We also displayed additional information given by the software which includes the Wilk Shapiro test (automatically performed) to check the normal distribution of the presented data in the graph, a p value of > 0.05 accept the null hypothesis of normally distributed data ( which was the differences  plotted against the means in this study).   

Indeed we changed our conclusion since very wide limited of agreements existed at  90% which over ruled even a partial  interchangeability according to our initial hypothesis. 

 We also  carefully analyzed results and verified distribution of all time points of comparison. We are grateful for your advice, we hope we relieved much of your justified  concern toward statistical issues. 

Second, for the correct application of B&A analysis, I suggest reading the following articles: https://pubmed.ncbi.nlm.nih.gov/26110027/ and 10.1016/j.tjem.2018.09.001.

As a consequence, for the lack of statistical analysis, the present article requires major revision.

We carefully read the suggested reference , thank you 

Reviewer 3 Report

Dear authors, 

Thanks for submitting your work to the journal. You describe a prospective study comparing the Datex neuromuscular transmission (NMT) kinemyography (NMTK) device with the TOFscan (TS) accelerometer. Patients (32) received, during anaesthesia, 0.5 mg/kg of atracurium or 0.6
mg/kg of rocuronium (age: 38–83 years, ASA: I–III). Bland and Altman analyses showed a bias (mean) of -59 seconds for onset [111; -130 sec] and 0 minute with an upper/lower limit of agreement of [-21.2; 21.2 sec]
during recovery. You conclude that the TS accelerometer might replace the NMTK during the last stage of recovery since calibration is not needed. 

The work is well done and well presented. 

Minor comments:

  • The abstract mention a bias of 0 minutes for recovery. How many seconds?
  • Methods: You should describe better the ethics approval: Who, where, what number, any registration on any website? Why in 2013 with a study conducted in 2018?
  • sugammadex without a capitalised s.
  • Table 1: Please present exact p-value. Alternatively, you may calculate the 95%CI and state in the legend that none of the p-value reach statistical significance (with defining the threshold for).

Author Response

Reviewer 3

Thanks for submitting your work to the journal. You describe a prospective study comparing the Datex neuromuscular transmission (NMT) kinemyography (NMTK) device with the TOFscan (TS) accelerometer. Patients (32) received, during anaesthesia, 0.5 mg/kg of atracurium or 0.6

mg/kg of rocuronium (age: 38–83 years, ASA: I–III). Bland and Altman analyses showed a bias (mean) of -59 seconds for onset [111; -130 sec] and 0 minute with an upper/lower limit of agreement of [-21.2; 21.2 sec]

during recovery. You conclude that the TS accelerometer might replace the NMTK during the last stage of recovery since calibration is not needed. 

The work is well done and well presented. 

Minor comments:

  • The abstract mention a bias of 0 minutes for recovery. How many seconds?

We changed the bias in sec and its around 36 sec

  • Methods: You should describe better the ethics approval: 
  • We added additional information about the ethical approval including the registration number , however this was not a trial  and was registered as a standard care research  protocol which are two different entities.  since no change in our practice was performed and both devices had E.C Autorization to use on patients,  and  consequently not registered in a specific  site. We also  delivered all documents to the editor before the reviewing began including EC autorization to use on patients for TOF scan. 
  • Who, where, what number, any registration on any website?
  •  The registration number as a research related to standard care is avis n° 2013 A00967 which is now added in the method section 
  • Why in 2013 with a study conducted in 2018?
  • The  study was approved by ethical committee and the gustaveroussy  IRB, there was no online registration. ,We first assessed the tof scan in comparison to tof watch  after publishing this study (3) , we had an opportunity to evaluate the toff cuff , we  therefore focused on tof cuff  for a certain time since it was a newer device until the study for the tofcuff was published (4) , However we continued also to acquire data for the present study , we are a very small team of anesthesiologists having this topic (neuromuscular monitoring)  in our department  of anesthesiology,   partly explaining the delay in the in publication of the second phase of assessment of TOF scan.

We respectfully do not think these explanations are relevant to be part of the manuscript since there is no scientific advantage on publishing the delay in publishing.

  • sugammadex without a capitalised s. We changed to sugammadex.
  • Table 1: Please present exact p-value. Alternatively, you may calculate the 95%CI and state in the legend that none of the p-value reach statistical significance (with defining the threshold for).
  • We now provided p values using wilcoxon test for paired data or studetnt t test for normally distributed data indeed  some of the distribution were not normal but fortunately the new results are not  very different except for 90% tof recovery as the limit of agreement  for ratio between devices  is more than 20% which is the limit clinically acceptable that we set   therefore because of this wide limit of agreement visible  in the BA presentation do not permit for us to conclude in interchangeability for  the devices under the condition of this study.
  • Thank you for your suggestions 

Round 2

Reviewer 1 Report

I would like to thank the authors for carrying out a thoughtful review of the manuscript, which enhanced its quality.

The only remaining issue with this research project, which unfortunately cannot be addressed by the review, is its slight impact on clinical practice and its minor potential, as a seed project, to initiate future research projects. Otherwise, I strongly suggest that the authors bring out the interest of this research smartly and sharply. It would be of great added value if the reader could quickly identify, from the introduction that leads him to the research question, the answer to the question "so what? ".

The fact remains that the project is adequately carried out, and the manuscript is easy to read.

Other minor comments

Table 1: Correct 1131 (95-158), is it 131?

Data are presented as mean (min) ± SD or median (min) (25-75) Percentiles as appropriate with regard to distribution: Should be corrected: (sec)

In Table 1, the time is in seconds, but the caption indicates minutes. In Table 2, the caption indicates minutes. Is this the case? Please check this out.

I hope this will help.

Author Response

The only remaining issue with this research project, which unfortunately cannot be addressed by the review, is its slight impact on clinical practice and its minor potential, as a seed project, to initiate future research projects. Otherwise, I strongly suggest that the authors bring out the interest of this research smartly and sharply. It would be of great added value if the reader could quickly identify, from the introduction that leads him to the research question, the answer to the question "so what? ".

Thank you for your suggestion

We now clearly stated our primary objective

This study was designed to assess the interchangeability of these two devices (TS and NMTK). Both devices have  the same external sensor shape, which can be employed in other positions than the classic extended arm, and placed at the wrist to stimulate the peripheral nerve of the adductor pollicis muscle during the onset and recovery of neuromuscular blockade.

We also added more comments in the results and discussion section about the clinical implication

Clinical implication from the present study for the onset of neuromuscular blockade is waiting until 0 response after TOF stimulation is probably too long especially when tof scan is used  confirming old investigations that laryngeal muscles are paralyzed earlier than adductor pollicis (18).

 This study also implies caution  with swift decisions under deep neuromuscular blockade (less than  four responses)  as variability of the results increase with both devices

The fact remains that the project is adequately carried out, and the manuscript is easy to read.

Thank you for your kind regards.

Other minor comments

Table 1: Correct 1131 (95-158), is it 131?

We apologize, this error has now been corrected  thank you.

Data are presented as mean (min) ± SD or median (min) (25-75) Percentiles as appropriate with regard to distribution: Should be corrected: (sec)

We apologize, this error has now been corrected thank you.

In Table 1, the time is in seconds, but the caption indicates minutes. In Table 2, the caption indicates minutes. Is this the case? Please check this out.

yes the onset is in sec , however recovery which is very longer is in minutes

Reviewer 2 Report

Dear authors

The present study compared two different methods to analyse neuromuscular blocking (kinemyography vs accelerometry). However, you did not state which methods you considered as "reference method". This aspect is peculiar for the right comprehension of your statistical results, particularly B&A analysis. It could be interesting to know how the two methods analyse the same phenomenon at the same time and, as a consequence, establish accuracy (bias) and precision (standard deviation) and then the grade of under or overestimation of the "alternative" method.

I suggest to review this aspect and correct the paper according to your choice of the reference method.

The materials and methods section is clear and well-written.

Result section: In general, you should add an explanatory caption for very tables and figures. This issue is crucial for full-comprehension of your statistical results.

I noted an error in table 1, about the median "Time to T4/T1 < 70% (sec)" for NMTK: 1131 (95-158) sec. Please correct the value of the median. 

Second, in the B&A plot for the onset of paralysis (figure 1), on the y axis, you should write a label (i.e. mean the difference between methods), and the Regression line showed a negative inclination with the increase in Mean of Onset NMTK and Onset TScan, while this did not show in figure 2. This suggested that bias increased with the increase in time to reach full neuromuscular blocking. At the induction of general anaesthesia, I supposed that some factors delayed the development of the full neuromuscular block, such as difficult airway management, in detail, difficult ventilation. You could try to elaborate your B&A using percentual variation and not absolute values for the y-axis. This could help you establish if increasing bias was related to measuring methods or clinical contest (i.e. difficult airway management).

Author Response

The present study compared two different methods to analyse neuromuscular blocking (kinemyography vs accelerometry). However, you did not state which methods you considered as "reference method". This aspect is peculiar for the right comprehension of your statistical results, particularly B&A analysis. It could be interesting to know how the two methods analyse the same phenomenon at the same time and, as a consequence, establish accuracy (bias) and precision (standard deviation) and then the grade of under or overestimation of the "alternative" method.

I suggest to review this aspect and correct the paper according to your choice of the reference method.

Thank you for your comment , indeed the NMTK is  the older method but not a reference or a gold standard method in this field,  in fact we already evaluated this device a few years ago against the gold standard method which is the force transducer method (1) and we found again important limits of agreements, denying interchangeability  therefore we do not consider the NMTK as the reference method.  In addition  if  we  considered the NMTK as the gold standard or reference method the classic graphic presentation would be under criticism as some authors (2)  suggest to plot the difference against  the gold standard method only and  not the mean of the 2 devices which would have drawn different or erroneous conclusions. Therefore for the sake of clarity we decided to present the NMTK as the first method (not the reference method per se )  and the TS as the second method which in fact was  also automatically suggested by our  statistical software (Med Calc V15.4). All differences are now described  in relation with  the first method (NMTK)

Nevertheless for general information purpose we reperformed the BA presentation by plotting the difference  against  the NMTK as a reference method and  surprisingly  we did not find any difference in the Bias and LOA currently described by using the mean between the devices,  may be because of the nature of our data!  Of course if the reviewer prefer this presentation we can change it.

1) C Motamed 1, K Kirov, X Combes, P Duvaldestin

Comparison between the Datex-Ohmeda M-NMT module and a force-displacement transducer for monitoring neuromuscular blockade:  European Journal of  Anesthesiology 2003 Jun;20(6):467-9 DOI: 10.1017/s02650215030007

 2) Jan S Krouwer

Why Bland-Altman plots should use X, not (Y+X)/2 when X is a reference method, Statistics in  medicine.

2008 Feb 28;27(5):778-80? DOI: 10.1002/sim.3086

The materials and methods section is clear and well-written.

Thank you for your kind comments

Result section: In general, you should add an explanatory caption for very tables and figures. This issue is crucial for full-comprehension of your statistical results.

We added some phrases for tables and figures  thank you for your suggestion

I noted an error in table 1, about the median "Time to T4/T1 < 70% (sec)" for NMTK: 1131 (95-158) sec. Please correct the value of the median.

Yes, thank you  this has been corrected , we apologize  for this transcription error.

Second, in the B&A plot for the onset of paralysis (figure 1), on the y axis, you should write a label (i.e. mean the difference between methods), and the Regression line showed a negative inclination with the increase in Mean of Onset NMTK and Onset TScan, while this did not show in figure 2. This suggested that bias increased with the increase in time to reach full neuromuscular blocking. At the induction of general anaesthesia, I supposed that some factors delayed the development of the full neuromuscular block, such as difficult airway management, in detail, difficult ventilation. You could try to elaborate your B&A using percentual variation and not absolute values for the y-axis. This could help you establish if increasing bias was related to measuring methods or clinical contest (i.e. difficult airway management).

Thank you for your comment, we plotted percentual variation not absolute values for onset of blockades  for y axis as you suggested , in fact the wilk Dhpairo p value was <0.001 and did not pass the normality test , indeed it decreased the LOA lines but the distribution remained roughly the same visually with  values  overlapping several times the LOA lines, and we had also a regression line deviated from the axis.

We believe it is more appropriate to leave the traditional presentation as  readers are more familiar with the mean of the difference rather than percentage of variation which might bring confusion  however if the reviewer believe this presentation is fundamental we are ready to  change it pending the distribution would not be normal and would require log and subsequently backlog transformation.

For  explanation of the  regression line by definition we did not include those patients which we anticipated having difficult intubation  (we added this latter  comment in the method section thank you ) and we believe the wide differences that we observed is typical for  the late phase  of onset of neuromuscular blockade since onset after intubating dose of muscle relaxants (at least  2ED95) will yield deep neuromuscular blockade permitting tracheal intubation, and as results of recovery especially deep neuromuscular blockade demonstrated in table 2, significant difference appears also  between devices at this level of paralysis. Indeed this regression lines confirm the statistical difference observed in  comparison of deep neuromuscular block during early recovery and we mentioned it by adding a sentence in the results.

This manuscript is a resubmission of an earlier submission. The following is a list of the peer review reports and author responses from that submission.